# A Fast Image Guide Registration Supported by Single Direction Projected CBCT

**Jian Gong** [1] , **Kangjian He** [1] **, Lisiqi Xie** [1,2] **, Dan Xu** [1,*] **and Tao Yang** [2]

1 School of Information Science and Engineering, Yunnan University, Kunming 650500, China; Gon3d@live.com (J.G.); Hekj@ynu.edu.cn (K.H.); Shirleys.qi@hotmail.com (L.X.)
2 Yunnan Leg Technology Co., Ltd., Kunming 650041, China; tsinxin@126.com
* Correspondence: Danxu@ynu.edu.cn

**Abstract:** Image registration is an important research topic in medical image-guided therapy, which is dedicated to registering the high-dose imaging sequences with low-dose/faster means. Registering computer tomography (CT) scanning sequences with cone beam computer tomography (CBCT) scanning sequences is a typical application and has been widely used in CBCT-guided radiotherapy. The main problem is the difference in image clarity of these two image sequences. To solve this problem, for the single projection image sequence matching tasks encountered in medical practice, a novel local quality based curved section encoding strategy is proposed in this paper, which is called the high-quality curved section (HQCS). As an optimized cross-section regularly encoded along the sequence of image, this curved section could be used in order to solve the matching problem. Referencing the independent ground truth provided by medical image physicians, with an experiment combined with the four most widely used indicators used on image registration, matching performance of HQCS on CT/CBCT datasets was tested with varying clarity. Experimental results show that the proposed HQCS can register the CT/CBCT effectively and outperforms the commonly used methods. Specifically, the proposed HQCS has low time complexity and higher scalability, which indicates that the application enhanced the task of diagnosis.

**Keywords:** image registration; medical image processing; image guide; CT imaging

## 1. Introduction

As a widely used technology in in vivo examination, a computer tomography (CT) device collects a series of X-ray intensity signals which reflects the X-ray absorbability of tissues by scanning the patient's body, and then the program reconstructs an image sequence of the body tissue's cross section based on those intensity signals [1–3]. In order to achieve a better fineness of image quality, a larger radiation dose was used on the patient's body for a more complete data image, but this also causes more serious radiation damage on the patient. Contrary to that approach, the cone beam computer tomography (CBCT) imaging technology reduces the radiation dose by sacrificing the details of soft tissue due to the low absorption rate of bone. CBCT is an optimized means for skeletal tissue imaging [4,5]. By replacing a full angle sustained scan by the short-time single direction X-ray projection, the exposure time of a patient in the radiation environment will be significantly shortened during CBCT imaging, and so is the damage from radiation. However, the shortcoming of CBCT is also obvious, and the low clarity of soft tissue makes the image look different than with the CT.

CT image-guided positioning programs are very common in radiotherapeutics. The doctor needs quite a long time to view the CT image and make the diagnosis, and during the most important job, which is sketching the target region for the therapeutist, patients are usually allowed to freely move [6–9]. When the patient goes back to bed, the body posture may be slightly offset. To avoid radiation damage to tissues outside the target

region, another CT scan is needed to check the posture changes and guide the therapeutist in repositioning the target region in the current body posture.

However, this is unlike the medical image matching problems which have complete data on three direction projections. In this paper, the data given for registration are two sequences of images from CT and CBCT scanning, but both are only projected in the spine section direction. An additional condition is that interlamellar spacing of adjacent image slices is constant in the same image sequence, but for most situations the interlamellar spacing is unknown [10–13]. In this context, the algorithm is required to find the best matching of two sequences, including the corresponding image slices between two sequences, and the alignment of two corresponding images.

To solve the insufficient registration accuracy caused by low quality of medical images in image sequences, we proposed a high-quality curved section (HQCS) based registration method. Our main contributions can be summarized as follows. Firstly, we propose a superimposition match-based method to reduce the effect of images with detail missing in the sequence on registration accuracy. Then, we propose a HQCS based method to determine the optimized cross-section for sequence registration, which effectively improves the registration accuracy.

The rest of this paper is organized as follows. Section 2 introduces the research background and data statement. Section 3 describes the proposed HQCS based registration method. Section 4 provides the experimental results and the relevant analysis. Section 5 draws the conclusion on the effectiveness of the HQCS approach and proposes future work using this technique.

## 2. Background and Related Work

### 2.1. Background

As described above, this paper is focused on a special sub-problem in medical imaging. The solution proposed attempts to solve two categories within the medical imaging problem. The first category is to reduce the time cost and radiation injury by registration of the CBCT image with a CT image; the second goal is to help the non-standardized medical serves, for example, the outmoded data process, or the multilevel authorization of medical image for consultation.

The pretreated data were provided from medical imaging professionals; to preserve patient privacy [14,15], all the identity information was removed. As shown in Figure 1, the image slices have been ordered from head to hip in spatial relation. In the same sequence, the interslice spacing is fixed but the exact value is unknown.

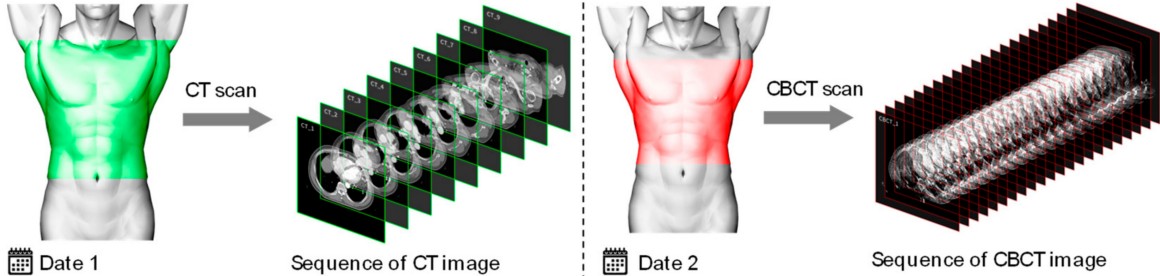

**Figure 1.** Two sequences of CT and CBCT image are sampled from the same patient, but the scan time is different.

The problem addressed in this paper is developing a method to look for the best corresponding CT and CBCT sequence; more specifically, this method has to determine the corresponding relationship of the image between two sequences (As shown in Figure 2). It was verified that the existing algorithms are competent for the task of matching single images. The main difficulty of those existing algorithms with our study is the inability to effectively determine the correspondence between sequences.

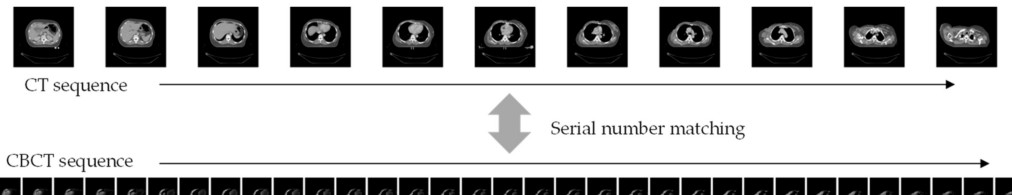

**Figure 2.** The slice order in sequence is given; the slice correspondence relationship is required based on image similarity.

The sequence of CT image slices is referred to as a CT sequence, or as a *CBCT* sequence, and the nth picture in a CT sequence simply CT_*n*; for example, in one of the datasets, the 15th CT image and 60th CBCT image are a correspondent pair which is a ground truth given by a medical imaging expert. This standard reference will be described below as "the pair *CT_15* and *CBCT_60*". If the pixels in an image sequence are seen as voxels in a 3D space *(x, y, z)* (shown in Figure 3), then the image space is in the *(x, y)* plane, and the task of this paper's method, using the pixels value as reference, is to find the optimal alignment of the CT and CBCT slice sequence in each of the *x*, *y*, and *z* directions.

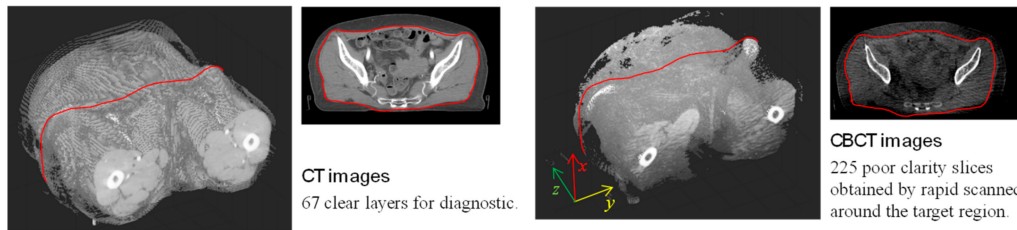

**Figure 3.** 3D voxels visualization of the CT and CBCT sequences, and the sample of image slice.

### 2.2. Related Work and New Challenges

The intuitive strategy is to consider the similarity of a pixel's value on the corresponding position and transform the scale and position of all CBCT pixels as a whole, then find the registration by searching the maximum/minimum target metrics. A few methods based on this strategy were improved previously for the medical image registration issue. Given the seriousness of the medical issues, the doctors prefer to choose an algorithm with more complete interpretability, such as the traditional registration method based on mutual information (*MI*) [16,17] or a structural similarity index measure (SSIM) [18].

Taking the mutual information index as an example to explain the specific difficulties encountered in our problem, the *MI* index could be calculated as Equation (1).

$$MI(R, F) = H(R) + H(F) - H(R, F) \tag{1}$$

$$H(X) = -\sum_{i=0}^{255} p(i) \cdot \log p(i) \tag{2}$$

$$H(X, Y) = -\sum_{i,j} p_{XY}(i, j) \log p_{XY}(i, j) \tag{3}$$

The *i* and *j* indicate the gray level of pixels in image *X* and *Y*, respectively. The *MI* model is focusing on the joint entropy of two images; if image *X* and *Y* were very similar, the $H(X, Y)$ would be close to 1. Conversely, the least value of $H(X, Y)$ is 0 when the two images are completely different.

The first difficulty for *MI* based registration algorithms is the disparity of image quality. The CBCT image is reconstructed from an incomplete signal, which caused the CBCT image to appear to be severely damaged. Figure 4 shows the result by a *MI* based registration method. It was mentioned above that *CT_15* and *CBCT_60* are a ground truth pair which the medical imaging expert would expect, but after it traversed over the entire *CBCT* sequence, the *MI* index shows that the response peak occurred at *H(CT_15, CBCT_56)* and

*H*(*CT_15*, *CBCT_64*). Repeated experiments show that the reason why the value of *H*(*CT_15*, *CBCT_60*) is wandering at the bottom valley is that the clarity of *CBCT_60* is so terrible. Comparing this with *CT_15*, even the content is different, but *CBCT_56* still has a better integration than *CBCT_60*.

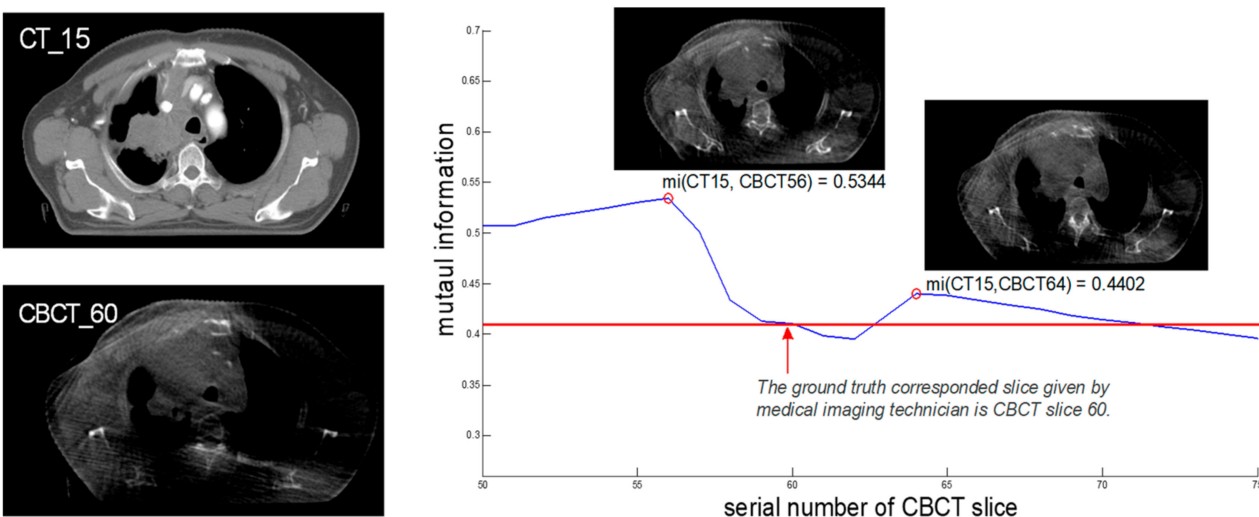

**Figure 4.** The *MI* response value of *CT_15* and each *CBCT* slice; the ground truth pair given by the expert has no significant advantage over the *MI* index.

The second challenge of traditional registration is the high complexity. Due to the high quality of image sequences collected by the new existing imaging technologies, the traditional registration method based on traversal matching will be time-consuming.

The last challenge is that the layer interval of sequence is fixed, which means that an integer correspondence relation of two sequences is usually not rigorous. In other words, if the interslice spacing of two adjacent image slices is 5 mm, this means that all the interslice spacing in this sequence is the integer times 5 mm. Assuming the interslice spacing of the other sequence is 1 mm, it appears that these were integer-fold relations. However, reality does not ensure that the two scans start with an integer multiple of 1 mm; in fact, it is quite often that the offset distance of the start position in two scans is uncertain. In most cases, the best corresponding position of a CT image in the *CBCT* sequence is somewhere between two *CBCT* slices, but not a point at one specific slice (as Figure 5 shows).

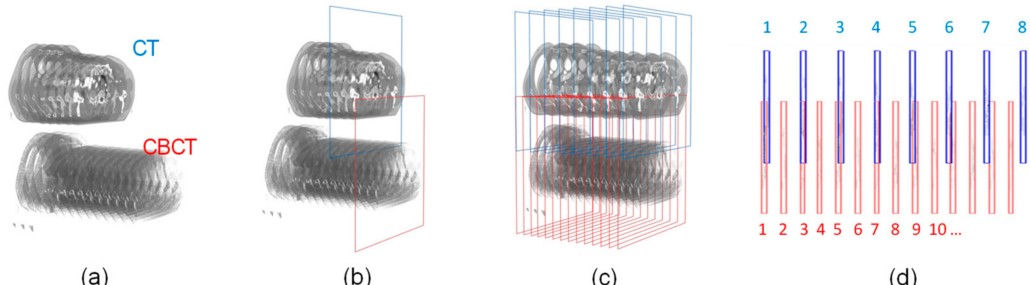

**Figure 5.** A possible situation of registration: (**a**) two sequences need to match; (**b**) the matched sequence; (**c**) cut out the overlapping segment; (**d**) the result of sequence matching—most of the *CBCT* slices (red color) have no aligned CT slice.

Thus, an integer correspondence relation of two sequences is usually not rigorous, and matching a strategy one by one may increase this error. The major single index-based registration algorithms currently in use have the above defects.

To solve the above problems, we proposed a high-quality curved section-based registration method for CT and *CBCT* sequences.

## 3. The Proposed HQCS-Based Registration Method

The proposed high-quality curved section-based registration method is shown in Figure 6, which mainly contains of three parts: superposition math, high-quality curved section (HQCS), and registration. The main strategy of our work is trying to reduce the impact of the low clarity area in *CBCT* image. To achieve this goal, the first contribution of our method (part I) is reducing the detailed proportion of a single image by superimposition match. The second contribution of our method (part II) is by calculating the local response of *CBCT* clarity. We build a high-quality grayscale feature with clear pixels, which is named as the high-quality curved section (HQCS). By HQCS, the obtained cross-section effectively bypasses the area where the *CBCT* image is severely inconsistent with the CT, and only retains the localities from which it is easy to find the corresponding CT image. Finally, the two sequences are registered by their HQCS map in part III.

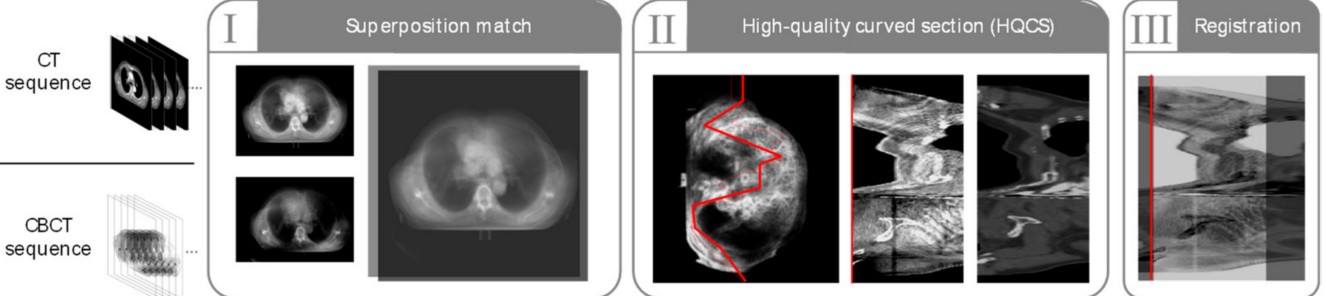

**Figure 6.** From left to right is the flow chart of our method. The far left is two input image sequences; part I is the process of the superposition match descripted in Section 3.1. In part II, a novel high-quality curved section (HQCS) is adopted, which is described in Section 3.2. Part III is the result of registration of two HQCSs composed in part II.

### 3.1. Superimposition Match

3.1.1. Alignment on Projection Plane

Assume a cross-section is encoded by images in the sequence and that it is description of some feature distributed in the *z*-axis direction, then the matching problem could be expected to be transformed into cross-section matching. This cross-section should possess three important properties:

**Property 1:** The cross-section is quantifiable, and always could be encoded by the images in sequence;

**Property 2:** The cross-section could be inverse mapped back to the order of image slices;

**Property 3:** The cross-section should be able to reflect a variation process occurring in the *z*-axis direction, and this process should be based on the same type of features existing in both the CT and *CBCT* sequence.

The first property makes sure a cross-section could be encoded from data already available; by the second property, the registered cross-section could be mapped back to the slice order in each sequence, then the corresponding relation is known. The last property is a most important one, as it ensures the cross-sections of two sequences could be used in a feature-based registration. For example, extract a column of pixels at the same position from each image in ordered image layers, and a cross-section conforming to the above properties could be formed by arranging the columns of pixels by this order.

A cross-section of the CT image sequence could be drawn by taking a straight line crossing the tissue, extracting the pixels along this line on all the images, and arranging those pixels column by column tightly, making sure the order of the pixel column is consistent with the image in the sequence so it is an example of the cross-section of a CT sequence that could be used for registration. This simple cross-section already has two of three properties mentioned above: the encoding method is simple and robust, and the order of the pixel column is unaltered; it is very easy to inverse map to the order number of

the image in the sequence. Additionally, this cross-section is also one of the CT slices in $(x, z)$ projection plane (a more detailed elaboration is in Section 4 of this paper). Of course, this could reflect the change in the structure in the z direction. However, before using this cross-section, two sequences should be aligned in the *(x, y)* projection plane. As Figure 7 shows, without alignment, the same straight line may cut through different parts of tissue, and it could lead to the two cross-sections representing a different feature.

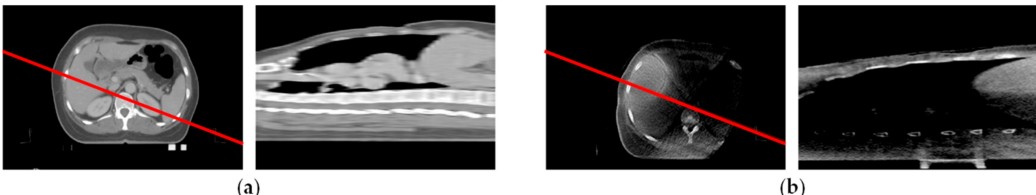

(a)                                                    (b)

**Figure 7.** The section of two sequences cut off by the same straight line, (**a**) the diagram of CT section line, and the cross-section rebuilt by the section line; (**b**) the cross-section of *CBCT* cut off by the same section line.

Therefore, the problem returns back to the matching of the multiple images. To overcome the dilemma outlined in the previous section, this paper proposes an overlapping matching method.

### 3.1.2. Superimposition Match

In Section 2.2, it is outlined that it is easy for the common feature-based registration methods to be misleading due to structural damage in *CBCT* images, thus providing the incorrect corresponding image. Without a reliable method to determine image correspondence, however, it is impossible to align the sequences by one to one matching. To illustrate the magnitude of the error, refer to the situation shown in Table 1. If trying to match *CT_15* with *CBCT_56*, the alignment transform parameter $T$ is (19,11), and the transform parameter to *CBCT_60* is $T_s$ = (18,10). Converting to a realistic scale, there is a risk of bringing a $(\sqrt{4^2 + 2^2 + 1} =)$ 4.5 mm error into an alignment parameter.

**Table 1.** Result of $(x, y)$ plane matching.

| | Ground Truth<br>*CT_15, CBCT_60* | Local Peak in Figure 4.<br>*CT_15, CBCT_56* | *CT_15, CBCT_64* | Our Method<br>Superposition |
|---|---|---|---|---|
| CT | | | | |
| *CBCT* | | | | |
| Registration | | | | |
| Transformation matrix $T$ | $\begin{vmatrix} 20 & 0 \\ 0 & 12 \end{vmatrix}$ | $\begin{vmatrix} 19 & 0 \\ 0 & 11 \end{vmatrix}$ | $\begin{vmatrix} 18 & 0 \\ 0 & 10 \end{vmatrix}$ | $\begin{vmatrix} 20 & 0 \\ 0 & 12 \end{vmatrix}$ |

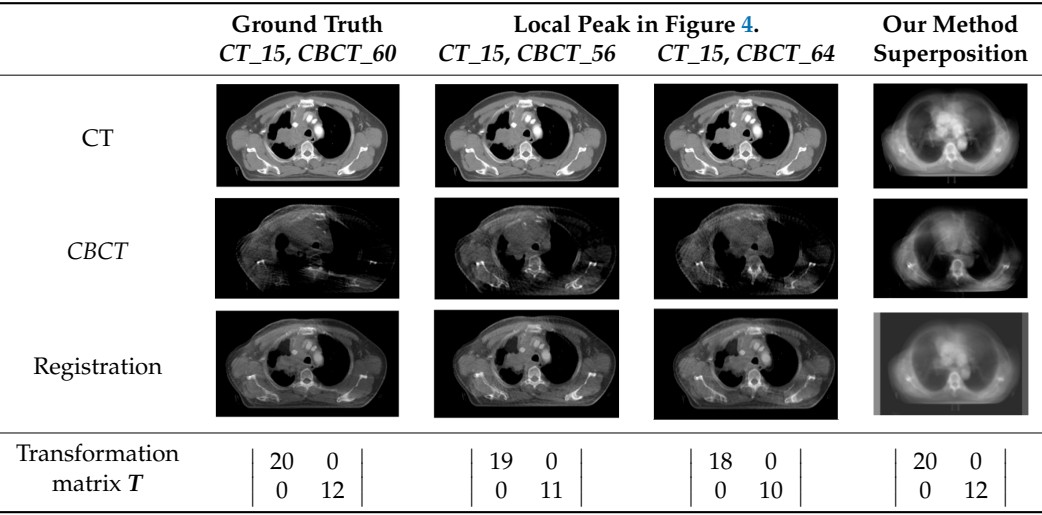

A superposition match ignores the clarity difference between each single image by considering all the data jointly. There might be several incidents of irradiation during a radiotherapeutics treatment; for the stage of CT/CBCT scan and radiotherapy, the patient has to try their best to hold a same body posture. This operation was designed to make the target locating easier, and in most cases, there is a customized pose container made for each patient to help the patient repeat body posture for each irradiation. This provides an

advantage from the global perspective of the image sequence, that the tissue on the image has already been set in the same shape, and thus a large range of affine transformation is unnecessary. Additionally, considering that the difference in the clarity is larger than the content details, the clarity disparities lead to the wrong peak value, as in Figure 4. Therefore, the impact of an overall profile needs to be enhanced intentionally in the matching process, while the clarity disparities are neglected properly.

Based on this view, we proposed the following superposition alignment method: overlay all the images in the CT sequence into a 2D matrix CT_Sp (Sp means Superposition) and overlay all *CBCT* images into a 2D matrix CB_Sp, then match these two matrices in the *(x, y)* plane to calculate alignment parameter *T* and the resultant registration (result is shown in last column of Table 1). In the superposition alignment method, neither image clarity nor detail difference do anything in the matching process directly, however, the whole contour formed by them is the real reference of the matching index.

### 3.1.3. Reconstruction of CT image

It is worth noting that, in Figure 5, that the scan distance of each sequence is inconsistent. How the extra part of the CT sequence influences a superposition alignment needs to be considered. Through the segmentation experiments, we found that the main contribution to registration comes from the areas of the spine. For a better demonstration, here is a brief tutorial about the CT reconstruction method in medical imaging. The fundamentals of CT reconstruction are that the absorption capacity of the material to X-ray is different according to its physical properties; usually the soft tissues have a higher absorption rate than bones, which are hard and which have small holes in volume, and therefore the energy loss is less when X-rays pass through bones relative to the X-ray decay being more severe when passing through areas where the soft tissue is thick. The reconstruction model of CT imaging was designed based on this law. The X-ray generator and receiver are positioned on opposite sides of body; the X-rays passing through the body hit a receiver which outputs the intensity of X-ray energy as the device revolves around the body, collecting a series of X-ray intensity signals from various angles. The Radon Transform is a widely used model to describe this imaging process [19,20], which can be described as:

$$R(s, \alpha) = \iint_{R^2} f(x,y)\delta(x\cos\alpha + y\sin\alpha - s)dxdy \tag{4}$$

where $f(x,y)$ is the compact support in $R^2$, which is the body torso in CT imaging. The $\delta(\mathbf{r})$ is the Dirac function, and the integrals proceed along the straight line $x\cos\alpha + y\sin\alpha = s$, which is the projection line in a CT scan. Seeking the Fourier transform of $R(s, \alpha)$,

$$
\begin{aligned}
F[R(s, \alpha)] &= \iiint_{R^2} f(x,y)\delta(x\cos\alpha + y\sin\alpha - s)e^{-j\omega s}dxdyds \\
&= \iint_{R^2} f(x,y)e^{-j\omega(x\cos\alpha + y\sin\alpha)}\left[\int \delta(x\cos\alpha + y\sin\alpha - s)e^{-j\omega(x\cos\alpha + y\sin\alpha - s)}ds\right]dxdy
\end{aligned}
\tag{5}
$$

where $F[f(x,y)]$ is the Fourier transform of $f(x,y)$. The Fourier transform result of the Dirac function is a constant, so Equation (5) could be reduced to:

$$F[R(s, \alpha)] = \iint_{R^2} f(x,y)^{-j\omega(x\cos\alpha + y\sin\alpha)}dxdy \tag{6}$$

The definition of the Fourier transform allows for the following:

$$F[R(s, \alpha)] = F[f(\omega\cos\alpha, \omega\sin\alpha)] \tag{7}$$

Judging from Equation (7), assume $f(x, y)$ is in the form of the X-ray absorbability of the scan area and the frequency domain of the scan area is same as the frequency domain of the series of X-ray intensity signals sampled by the CT device. Considering that the sample angle of each intensity is already recorded during the scan process, the time domain of this series of intensity is re-buildable. Overlying the Fourier transformed signal along the angle of the irradiation projection, the superimposition of all transformed signals is the X-ray absorbability distribution of the scan area. The image shaded by this absorbability from

weak to strong is the CT image often used in the hospital. This kind of reconstruction based on back-projection is commonly referred to as iRadon Transform.

In industrial practice, the superposition is along the angles of the signal scanned, however, the pixels of the digital image are arranged at right angles. To reduce the periphery shadow in sub-pixel superimposition, another FFT process is needed after the iRadon Transform [21]. The value of pixels in a CT image represents the local pass ability of an X-ray. If the local absorption capacity was strong, the residual energy of an X-ray is low and the region is darker; relatedly, the light region correspondents with the low absorption capacity. The pixel value of a CT image is not about the optical reflection, rather it expresses the local X-ray absorption characteristic of tissues.

According to the fundamentals of the CT reconstruction method, it is known that the CT image is formed by accumulation of a local X-ray passing rate. The pixel in the CT image represents a local X-ray's absorption capacity, thus, in the previous section, the simple cross-section was viewed as a CT image in the projection plane $(x, z)$ and is in line with this fundamental. Furthermore, it indicates that the overlapping is a meaningful operation in medical image processing.

Due to the fact that the patient is usually asked to lay flat along the irradiation direction vertically, observing at the direction of the projection plane of the CT images, the center of spine is basically coincident, therefore, a small divergence in the number of overlying images only makes the statistical frequency of the signal different, and the influence on the central axis matching the spine is insignificant, which as shown in Figure 8.

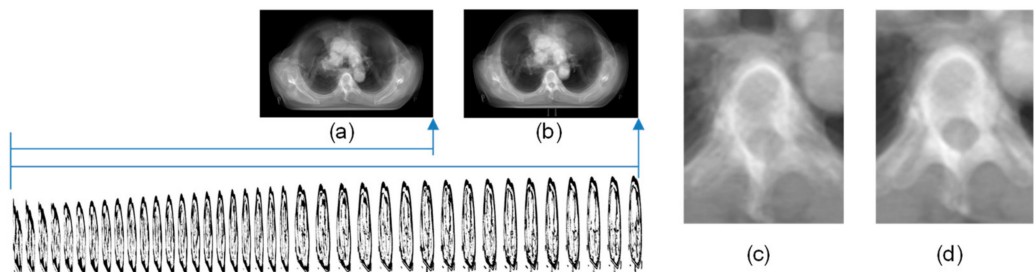

**Figure 8.** The test of superimposition with different layer number (**a**) is the superimposition image of 30 slices overlaying or (**b**) is the superimposition of 40 slices overlaying. (**c**,**d**) is the partial enlargement of (**a**,**b**).

### 3.2. High-Quality Curved Section

### 3.2.1. Motivation and Principles of HQCS

The simple cross-section in Section 3.1.1 could be viewed as one of the CT images projected in the plane $(x, z)$ which has been elaborated on above, but in the practice of our work, a simple cross-section usually does not work. In most cases, either a simple cross-section or the superimposition of the simple cross-sections are unable to lead to a desired result of a z-axis registration. One of the major reasons is shown in Table 2, the contour in the projection plane $(x, z)$ or $(y, z)$ is not as sophisticated as in the $(x, y)$ projection plane; without the rate data of the stretching ratio, it is difficult to determine the positional relationship of the two short gently varying curves. The scale parameter of layer spacing is not included in most data encountered in this paper, so the stretching ratio of the contour cannot be calculated directly.

The fundamentals of *CBCT* imaging are the same as the CT; the difference between them is the sampling process during a scan. The *CBCT* is a special X-ray-based imaging technology. By a dedicatedly designed X-ray generator, the X-ray irradiation in a pyramidalis volume is mainly designed for dental imaging or for reducing the radiation damage in head scan, and the exposure dose is lower than the annular CT device [22–24]. For the migrated *CBCT* which is applied to a body scan, the scan device only moves along the single direction of the z-axis without the rotation around the patient, so it is impossible to reconstruct an image with the same level of quality as the annular CT imaging. As Figure 9

shows, in some *CBCT* body scans, there are some areas where the data are obviously missing. An image of this quality is insufficient for diagnosis, but medical experts still use the advantage of a low exposure dose of the *CBCT* scan by image guide technology.

**Table 2.** Registration based on simple cross section.

| | Vertical Plane Section | Horizontal Plane Section | Horizontal Superposition |
|---|---|---|---|
| CT | | | |
| *CBCT* | | | |
| Registration | | | |
| Spacing rate, ground truth = 5 | 6.0333 | 4.7 | Registration failed |

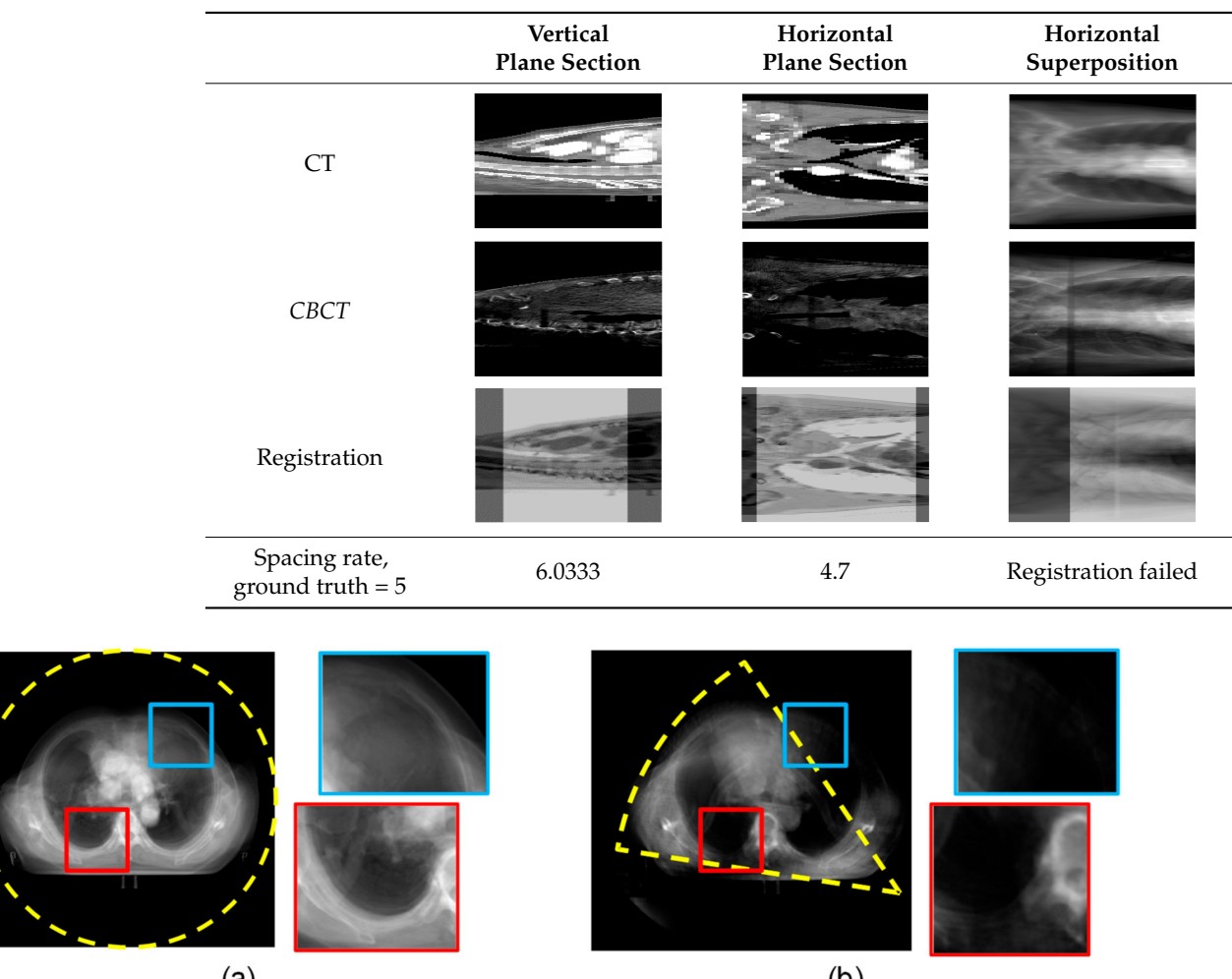

(a)       (b)

**Figure 9.** (**a**,**b**) are the superposition of the CT and the *CBCT* sequences, the yellow dashed circle indicates the region with an abundant projection signal, and the rectangular boxes indicate two examples of missing data areas in the *CBCT* scan.

Considering that the fundamentals of *CBCT* and CT are the same, a pixel in a *CBCT* image has the same meaning as in the CT, even though the CT has accumulated added signal by more sampling angles; however, the value of the reconstructed image needs to be normalized in 0 to 255, the same as the range of a *CBCT* image. Therefore, for the area with better image quality in the *CBCT* sequence, the content is very similar to the CT image.

### 3.2.2. High-Quality Curved Section for Registration

Assuming there is an aligned pair of the CT and the *CBCT* images (by the matching method demonstrated above), take the CT image as a reference, then normalize the *CBCT* image with the same dynamic range as the CT image; ideally, the fidelity of any pixel $i(x, y)$ in a *CBCT* image could be estimated to refer to the corresponding pixel $ict(x, y)$ in the CT image.

Based on the idea of avoiding being included with the error, we proposed an evaluation model based on local standard deviation:

According to the CT and *CBCT* reconstruction method, the projection angle of CT scan is a completely covered target area, but the *CBCT* is not, so the upper bound of *CBCT* signal accumulation will not exceed the CT's upper bound, but the lower bound for both is 0. This means the original dynamic range of the *CBCT* image is smaller than the CT's; in the normalization procedure, these two different ranges are linearly mapped to be [0, 255]. The CT with higher upper bound will press the low intensity interval into a more narrow grayscale interval, making the low intensity interval in the *CBCT* image maintain a more significant frequency change than the CT image.

The quality evaluation is based on the reference of a CT image for the final evaluation of the low-intensity regions mentioned above. Hence, we use the local information quantity of a CT image to limit the impact by low-intensity regions; Equation (8) was designed as an evaluation for the local information quantity based on a CT superposition image.

$$E(I_{ct}) = H\left[\iint ct(x_0 - x, y_0 - y)g(x, y)dxdy\right] \tag{8}$$

$$g(x, y) = \exp\left[-\left(\frac{(x - x_0)^2}{2\delta_x^2} - \frac{(y - y_0)^2}{2\delta_y^2}\right)\right] \tag{9}$$

The $ct(x, y)$ is a signal function designed to return the pixel value at coordinate $(x, y)$ in a CT superposition image. The variable $g(x, y)$ is a two-dimensional Gaussian kernel function in Equation (9), the variable $H(x)$ is the two-dimensional information quantity computation function given in Equation (2).

$$L(I_{ct}, I_{cbct}) = \frac{\sum \min(I_{ct}, I_{cbct})}{\sum I_{ct}} \tag{10}$$

Equation (8) is designed to emphasize the weight of high-frequency components in the CT image, and Equation (10) is designed to reduce the impact of a low grayscale interval in the *CBCT* image. For the areas where the normalized *CBCT* image retains a larger intensity than the CT image, we are not going to judge whether the larger intensity is caused by the principle mentioned above, and the reconstruction procedure reveals that the CT imaging has a more completed signal sampling than the *CBCT*, so the CT data are trusted preferentially in this work. In Equation (10), $I_{ct}$ is the discrete express of $ct(x, y)$. This function takes $I_{ct}$ as a reference to estimate the reliability of pixel in the *CBCT* image. Usually, $i_{cbct}$ is smaller than $I_{ct}$, the value of $I_{cbct}$ is closer to $I_{ct}$, the $I_{cbct}$ is more reliable, and the response of function is larger. For the case of the area where $I_{cbct}$ is larger than $I_{ct}$, $I_{cbct}$ was supposed to be retained in our method. We analyzed this case in Equation (8) and returned an impact limited function based on the local information quantity.

Combining the Equations (8) and (10), the local quality evolution function for *CBCT* image is proposed as:

$$\begin{aligned} Q(I_{ct}, I_{cbct}) &= L(I_{ct}, I_{cbct})E(I_{ct}) \\ &= \frac{\sum \min(I_{ct}, I_{cbct})}{\sum I_{ct}} H[\sum\sum i_{ct} * g(i)] \end{aligned} \tag{11}$$

where $*$ denotes the convolution operator. This evaluation model includes the discrete format of Equation (8), and $i$ is a specific pixel value in the image coordinate.

This equation was designed to evaluate the local similarity of the *CBCT* image referred to the CT image. The output of $Q(I_{ct}, I_{cbct})$ is an evaluation map, which is used to reflect the local clarity of the *CBCT* image and a guide to how the program extracts the pixel with a high quality of detail, to build a cross-section with the sharp and continuous contour of the tissues. To this purpose, we curved the cross-section to fit the distribution of peak valued areas in the evaluation map, so since the optimized cross-section is no more than a flat plane cutting in the object, it should be a surface curling in the object volume.

Due to this, an optimized cross-section is formed based on a high-clarity pixel screening strategy, and the pixels of this cross-section are distributed along a curved surface spatially, so the flattened image of this curved cross-section is named "high-quality curved

section" (HQCS). Figure 10 shows an example of HQCS. The HQCS is an image, which is supposed to be used in a sequence registration problem as a spatial feature. It can be seen that the HQCS obtained by us containing significant information in the stereo space, which is more effective and accurate than a randomly processed cross section. In the experimental section, we demonstrated that the registration results using HQCS outperformed the traversal registration-based method.

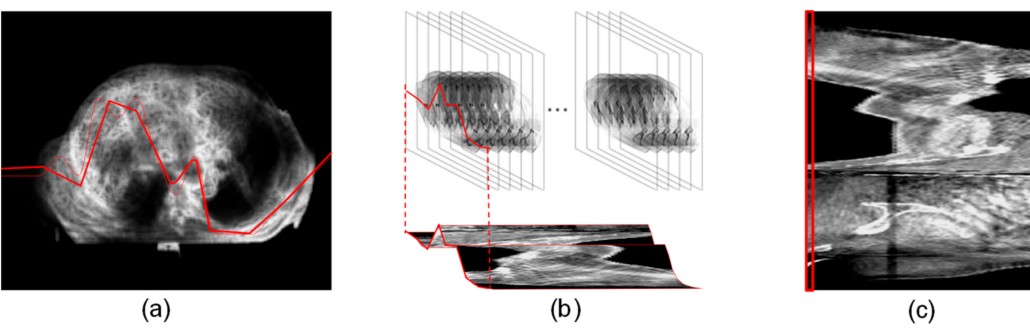

**Figure 10.** The diagram of a high-quality curved section (HQCS). (**a**) is the evaluation map $Q(I_{ct}, I_{cbct})$ computed by Equation (11), in which the red line is a section curve fitted by the local peak values in evaluation map. (**b**) is the curved section in the 3D view of the source medical image sequence. (**c**) is the HQCS used for registration in our work. It is a flattened curved section in the graph (**b**).

## 4. Experiments and Analysis

### 4.1. Experimental Setups

To evaluate the performance of the proposed HQCS, we have employed four image registration indexes commonly used for medical image registration, the mutual information (*MI*), Structural Similarity (SSIM), Standard Deviation (SD) [25], and normalized cross correlation (NCC) [26,27], and four datasets were chosen for an experiment. They cover two typical discrepancy situations in a *CBCT* image, which are the low clarity and an incomplete projection. The example images of the four datasets are given in Table 3. Each dataset has two sequences of gray level imaging, which is the CT image sequence and the *CBCT* image sequence. A special condition of the experiment dataset is that the layer spacing is known, the layer spacing of all the CT sequences is 5 mm, and the *CBCT* sequence is 1 mm.

**Table 3.** The example image of four datasets used for verification experiment. # denotes the amount of layers of image sequences.

|  | Low Clarity | | Incomplete Projection | |
|---|---|---|---|---|
|  | Data set **a** | Data set **b** | Data set **c** | Data set **d** |
| *CT* sequence | #204 | #42 | #56 | #350 |
| *CBCT* sequence | #225 | #160 | #184 | #264 |

Moreover, the medical experts helped us to determine the corresponding layer of the first *CBCT* layer in the CT sequence. This correspondence is used as the ground truth for

estimation of the performance of our method. All the datasets are collected and produced by us and have been anonymized. Our experiments are performed using MATLAB codes on a PC with 2.3 GHz Intel Core i5 CPU and 8GB RAM.

### 4.2. Performance Analysis

We tested the datasets with four common registration indexes separately. For each index used in the experiment, two computational schemes are included: one is the traditional strategy which requests a traversal to find a pair of simple cross-sections with the large response of indexes, then a bidirectional stretch and a shift of simple cross-sections. When the index response achieved is maximum, the correspondence of the two sequences is the registration result. The second scheme is registration based on the HQCS process proposed in this paper. The first step is superimposing each sequence to calculate the evaluation map. Next, extract the pixels along the high-response areas in the evaluation map, and build the HQCS with the extracted pixels. Finally, match two HQCS by the index-based registration and output the result.

As mentioned above, the result of registration contains two parts: the layer spacing and the corresponding CT image to the first layer of *CBCT* sequence. The ratio of layer spacing is given in Table 4. The second part of result is given in Table 5 by the CT layer number which corresponds to the beginning of the *CBCT* sequence.

**Table 4.** The layer spacing ratio of *CBCT* to CT, ground truth = 5/1 (mm).

| Dataset | *MI* | | SSIM | | NCC | | SD | |
|---|---|---|---|---|---|---|---|---|
| | No HQCS | HQCS | No HQCS | HQCS | No HQCS | HQCS | No HQCS | HQCS |
| **a** | 4.87 | **5.046** | 5.08 | <u>**5.0**</u> | **4.74** | 4.73 | 5.046 | **5.015** |
| **b** | 5.2 | **4.97** | 5.15 | <u>**5.025**</u> | 4.7 | **4.90** | 6 | **5.5** |
| **c** | 5.08 | <u>**4.95**</u> | *Failed* | **4.93** | 4.58 | 4.57 | *Failed* | **4.90** |
| **d** | 5.95 | <u>**4.98**</u> | 6.75 | **5.35** | *Failed* | **4.8** | *Failed* | **5.67** |
| Avg deviation | 0.34 | <u>**0.0378**</u> | 0.858 | **0.113** | 0.328 | **0.21** | 0.523 | **0.321** |
| Time cost | 900 s–1200 s | **130 s–220 s** | 800 s–1400 s | **130 s–220 s** | 600 s–700 s | **42 s–180 s** | 600 s–700 s | **30 s–120 s** |

The bold data is the better one in the control group of the same index; underlining data is the best of the whole row, and the italics indicate the result is error.

**Table 5.** The sequence number of CT layer corresponding to the first *CBCT* image.

| Data\Ground Truth | *MI* | | SSIM | | NCC | | SD | |
|---|---|---|---|---|---|---|---|---|
| | No HQCS | HQCS | No HQCS | HQCS | No HQCS | HQCS | No HQCS | HQCS |
| **a**\13 | 7.18 | **12.88** | <u>**13.0**</u> | <u>**13.0**</u> | **12.01** | 11.61 | 13.08 | **12.96** |
| **b**\3 | 4.23 | **2.98** | 4.27 | <u>**3.18**</u> | **3.40** | 3.6923 | **3.83** | 4.18 |
| **c**\4 | <u>**3.94**</u> | 3.36 | *Failed* | **4.67** | **3.49** | **3.49** | *Failed* | **3.26** |
| **d**\6 | 8.74 | **5.02** | 9.70 | **6.35** | *Failed* | <u>**5.79**</u> | *Failed* | **6.87** |
| Avg deviation | 2.462 | **0.437** | 1.655 | <u>**0.302**</u> | **0.635** | 0.699 | **0.456** | 0.707 |
| Deviation distance (mm) | 12.31 | **2.18** | 8.27 | <u>**1.51**</u> | **3.17** | 3.49 | **2.28** | 3.53 |

The bold data is the better one in the control group of the same index; underlining data is the best of the whole row, and the italics indicate the result is error.

In Table 4, the datasets **a** to **d** are arranged vertically, and the table is divided into four major columns according to the four indicators; each indicator includes two columns of data, the left column is the test result by a traditional scheme and titled as "No HQCS", the right column is the result by the HQCS based registration and titled as "HQCS". The bold item is the better one and the underlining data are the best result of all the methods on the same dataset. The "*Failed*" item means the result seriously deviated from the ground truth.

From Table 4 it can be found that the traditional scheme without HQCS costs much more time than the registration scheme with HQCS. Due to the fact that the proposed HQCS is not built by searching, which means that it has lower time complexity, as shown in Table 4, especially for the incomplete *CBCT* imaging datasets, the proposed method shows outstanding advantages in processing time consumption.

The visualization of Table 4 is plotted in Figure 11. The bars representing the same dataset are colored by the same hue, and the HQCS one is deeper. The ground truth is marked by a red dotted line; the closer the height of the bar to the dotted line, the better the bars' data.

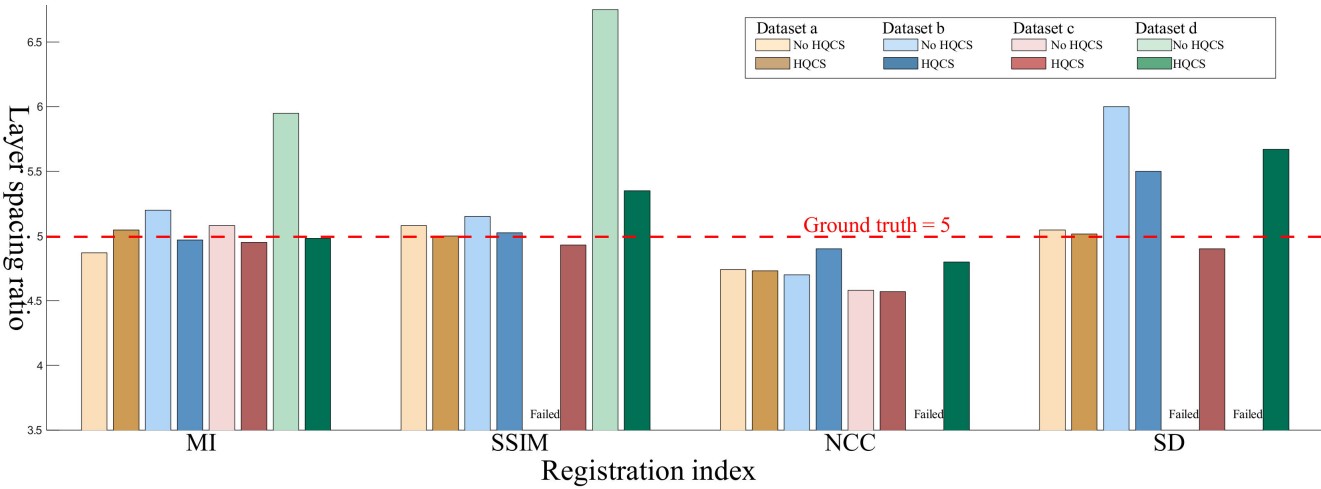

**Figure 11.** The figure of Table 1, red dotted line $y = 5$ is the ground truth marker for all data, each cluster of histograms represents the registration results of the index marked below.

More importantly, the registration accuracy without adopting HQCS is lower than the results we obtained by HQCS. In Table 5, we listed the registration accuracies for the registration scheme without HQCS and with HQCS. The dataset number and corresponding ground truth are given in the first column of each row. Followed by the experimental results. "a\13" represents the ground truth corresponding pairs of dataset A pair (*CT_13*, *CBCT_1*), and the second column shows the result of the traditional scheme without HQCS is (*CT_7.18*, *CBCT_1*). The third column of this row shows that our method obtained the registration result of 12.88 on dataset A by *MI* based method, which is closer to the ground truth than the traditional scheme, so this item is in bold. Additionally, the SSIM index provides the best registration result on dataset A (both of the HQCS based and the traditional schemes, the results are equal to the ground truth in percentile accuracy). As the best data of this row, these two items are marked out by underline.

As can be seen in Table 5, the registration accuracy is significantly improved both on *MI* and SSIM by adopting the HQCS to the registration scheme. At the same time, HQCS can effectively overcome the failure of registration, considering the fact, given in Figure 5, that the layer positions of two random scans are pretty hard to overlap accurately. An output of the positive real number with the deviation less than one layer is acceptable.

To further demonstrate the effectiveness of the proposed method, we also performed the comparison experiment with the intensity-based method [28]. The comparison results are given in Table 6. It can be found that the intensity-based registration method slightly outperforms the *MI* based method. However, by adopting the proposed HQCS, the registration accuracy based on *MI* is greatly improved.

Figure 12 shows the final registration results of the proposed HQCS method on dataset C. It can be found that the proposed method has registered the CT and *CBCT* sequences accurately. The visualization of Figure 12 also demonstrates the promising applications of our method in medical diagnosis.

**Table 6.** The comparison experiment with intensity based registration.

| Registration Method | Intensity Based (Single Image) | Intensity Based (The Sequence) | *MI* Based (No HQCS) | *MI* Base (HQCS) |
|---|---|---|---|---|
| Result Ground Truth: (*CT_15*, *CBCT_60*) | (*CT_16.7*, *CBCT_60*) | (*CT_14.26*, *CBCT_60*) | (*CT_15.76*, *CBCT_60*) | **(*CT_15.05*, *CBCT_60*)** |
| Visualization | | | | |

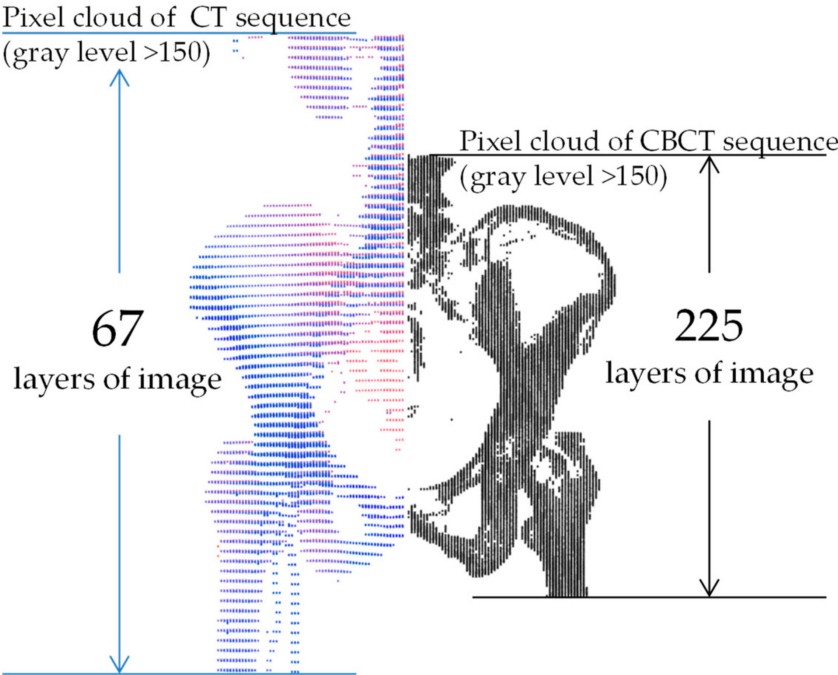

**Figure 12.** The planform of registered dataset **C**, by HQCS based registration method with *MI* index. The pixel cloud only retains the pixel value greater than 150.

## 5. Conclusions

In this paper, we proposed a novel high-quality curved section (HQCS) based registration method, which aims to register the CT and *CBCT* image sequences quickly with both high accuracy and robustness. The proposed HQCS based registration method mainly contains superposition math and a high-quality curved section (HQCS). Firstly, in the sequence on registration accuracy, we proposed a superimposition match-based method to reduce the effect of images with detail missing. Then, we proposed a HQCS based method to determine the optimized cross-section for sequence registration. Experimental results show that the proposed HQCS can register the CT/*CBCT* effectively. Specifically, the proposed HQCS has low time complexity and higher scalability, showing the application prospects in medical diagnosis. Next, ways to optimize the HQCS strategy combined with machine learning [29] and apply the registration results to disease detection will be considered in our future work.

**Author Contributions:** Conceptualization, J.G., K.H. and D.X.; methodology, J.G. and K.H.; software, J.G.; validation, K.H., T.Y. and L.X.; formal analysis, D.X. and T.Y.; investigation, L.X. and T.Y.; resources, T.Y.; data curation, T.Y. and L.X.; writing—original draft preparation, J.G.; writing—review and editing, K.H.; visualization, J.G.; supervision, D.X.; project administration, D.X.; funding acquisition, D.X. and L.X. All authors have read and agreed to the published version of the manuscript.

**Funding:** This research was supported in part by the National Natural Science Foundation of China under Grant No. 62162068 and Grant No. 61761049, in part by the Yunnan Province Ten Thousand Talents Program and Yunling Scholars Special Project under Grant YNWR-YLXZ-2018-022, and in part by the Yunnan Provincial Science and Technology Department–Yunnan University "Double First Class" Construction Joint Fund Project under Grant No. 2019FY003012.

**Institutional Review Board Statement:** Not applicable.

**Informed Consent Statement:** Not applicable.

**Data Availability Statement:** Not applicable.

**Conflicts of Interest:** The authors declare no conflict of interest.

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
