# Peer review of "A Fast Image Guide Registration Supported by Single Direction Projected CBCT"

_electronics, doi:10.3390/electronics11040645_

Round 1
Reviewer 1 Report
See the comments in the PDF file. The organization and presentation of the paper and figures have been greatly improved. However more attention needs to be placed on the english used to discuss your research. I went through the entire paper this time making changes I thought were needed. You should verify I have not changed the discussion you were trying to present. Your native english speaker did not do a very good job becuase there were many instances that needed to be addressed. The research however is solid. I'm looking forward to a better presentation of your work so it can be published.

Author Response
On behalf of all the co-authors, I would like to express my gratitude for the constructive remarks and useful suggestions, which has significantly raised the quality of our manuscript and has enable us to improve the manuscript. It must take up a lot of your time and attention, each of your comment has been accurately revised. We gratefully appreciate for your valuable and detail comments. All the changes are marked in red color in this version. Thank you again for your meticulous emendation.

Reviewer 2 Report
The presentation of this resubmitted version is much better than the previous ones.
To make clear how contribution of superposition match (not math) and HQCS to the overal performance, the authors should highlight in the Conclusion. In addition, since the Superposition match is one part of HQCS (as stated by the authors), it is not clear why HQCS includes superposition match and HQCS ?
The authors said the results of 12.88 on dataset is very close to the groundtruth, what is the main reason for this ?
Author Response
Thanks for your professional and careful review on our article, which are very helpful for us to improve our manuscript. You noticed the special narrative structure of superimposition match and the HQCS model, in this regard, we supplement the following explanation:
“The result of superimposition match is the prerequisite for the HQCS encode, but superimposition match is not the only way to provide this prerequisite. The reason why we present it as an independent part, because it just right coincides with the physical meaning of CT imaging. Besides, the method of superimposition match has the advantage of fast and reliable, it is a fine choice for our research background. But due to the main issue of our work is the HQCS based registration scheme, we didn't discuss further on superimposition match method, and we also looking forward to a better suitable method to solve the aforementioned preprocess of HQCS encode.”
For the second problem which about experiment result, we are sincerely grateful for your professional reminding, we realized that the statement of this part is not clear enough. We have added more descriptions about this part in the new version of manuscript(The details are attached).

Reviewer 3 Report
The article is revised as per suggestions. I have no objections to accepting and publishing it.
Author Response
Thanks. We really appreciate your previous comments and suggestions. Those comments are valuable and helpful for revising and improving our article. And we are very grateful for your positive evaluation of our work.
Round 2
Reviewer 1 Report
I am finally able to read through the entire paper with only a few rough spots (very few). Congratulations for making improvements which better allow the description your work to be closely followed. If I were yo I would read the entire paper one more time and I think you will catch the few instances (rough spots) I mentioned. After the this last read thru I think it is ready for publication.